# Investigation and Development of the Brushless and Magnetless Wound Field Synchronous Motor Drive System for Electric Vehicle Application

Yanhui Li , Yiwei Wang, Zhuoran Zhang * and Jincai Li

Jiangsu Key Laboratory of New Energy Generation and Power Conversion, Nanjing University of Aeronautics and Astronautics, Nanjing 211106, China
* Correspondence: apsc-zzr@nuaa.edu.cn

**Abstract:** In order to solve the problems of soaring costs and supply fluctuation in permanent magnet materials, this paper investigates and develops a magnetless wound field synchronous motor (WFSM) drive system for electric vehicle (EV) application. As the crucial drive component for EVs, the proposed WFSM in this paper has a two-stage structure which is described as the exciter and the main motor. The excitation characteristics of the exciter that provide power to the rotor field winding are emphatically analyzed. In addition, the discrete time domain armature current regulator and excitation current regulator are designed and analyzed for the high-performance WFSM drive system. A current coordinated control strategy for the full speed range is proposed to expand the constant power region. The experiment shows that the excitation characteristics and torque capability of the WFSM are consistent with the analysis, and proves that the WFSM is a potential solution for the magnetless motor for EV application.

**Keywords:** electric vehicle drive system; magnetless machine; wound field synchronous motor

## 1. Introduction

In light of the increasing concerns about environment pollution and a reduction in gas emissions, the environmental problems brought about by the transportation industry cannot be ignored. The automotive industry is focused on developing sustainable products and innovative strategies for gas emission reduction. The market share of pure electric vehicles (EV) that are powered by electric motors as the traction component is becoming larger. Considering that traction motors need to have high power density, efficiency, low torque ripple, and low noise and vibration, permanent magnet synchronous motors (PMSMs) are widely employed for EV traction [1,2].

However, because of the environmental disruption when mining the rare earth as well as serious price fluctuations caused by an imbalance between supply and demand [3], the requirement of reducing the use of the rare earth is growing in many industries. Thus, many alternatives for the PMSM such as the induction motor (IM), reluctance motor (RM), and wound field synchronous motor (WFSM) are being researched [4,5]. Instead of being excited by permanent magnets, field windings in the rotor are employed in the WFSM [6–8]. WFSMs offer several potential benefits and advantages including operation without permanent magnets. Drives based on the WFSM should be able to generate high torque at low speed by overloading both the armature and the field windings. In this way, the armature current is kept at acceptable values, avoiding excessive oversizing of the power electronic converter [9,10]. For high-speed operations, PMSMs only achieve flux-weakening control, whereas WFSMs achieve flux-weakening control and field-weakening control. Compared with IM and RM, the drive system based on WFSM is suitable to satisfy the high torque demand of heavy electric vehicles at standstill and low speed, as well as the constant power operation in a wider speed range. Moreover, by using high-frequency

signal injection in the field winding, sensorless control independent of the saliency ratio of the machine can be achieved [11].

Scholars have researched the scheme of the WFSM to replace the rare earth PMSM and have optimized their design. The flat wire hairpin winding is applied in the WFSM to reduce copper loss [12], and the design of core and winding parameters is optimized by the integrated evaluation method of mechanical, copper, and iron loss, which increases the power density of the WFSM by 8.5% [13]. In order to further improve the operating efficiency of the WFSM, it is proposed that inserting auxiliary permanent magnets in the WFSM can further increase the torque and reduce the armature current [14]. The high current density of the WFSM rotor windings leads to heat accumulation, and the traditional spiral stator water jacket cannot provide effective cooling for the rotor. Therefore, research reports on automatic transmission fluid spray cooling specially designed for the WFSM [15].

In 2013, two EV traction motors using WFSMs were introduced commercially. Renault is using a WFSM in their Renault Zoe EV [16]. However, the dc field in this motor is excited externally by using brushes. As a mature technology, the use of slip rings and brushes [17,18] is simple but generates undesirable dust and demands periodic maintenance, and the friction loss during continuous operation limits the further improvement in efficiency. In order to mitigate these issues, inductive coupling and capacitive coupling to transfer power to the rotor winding of a wound field machine have been reported recently [19,20].

This contactless excitation of rotor winding is a feasible solution. A contactless exciter operating in rotating transformer mode is investigated and developed to excite the rotor field winding in this paper. The field power is transferred through an air gap with no mechanical contact, and is rectified by the rotating rectifier to excite the rotor winding.

The WFSM has a higher degree of freedom (DOF) of controllability because field-weakening control can be achieved by controlling the field current and the armature current, whereas it introduces control complexity through the inherent physical cross-coupling between the direct and field axes. Most high-performance WFSM control solutions propose using separate PI current regulators on d, q, and field axes [21,22]. The excitation current coordination control method based on the constant voltage vector is proposed in this paper to minimize the complexity of the control system. All current regulators are designed in the discrete time domain and can be directly applied to digital control systems using micro control units.

## 2. Structure and Principle of the Proposed WFSM Drive System

Figure 1 depicts the basic two-stage structure of the proposed WFSM which consists of the exciter and the main motor. The two parts are electrically connected by a rotating rectifier composed of diodes. The main motor and exciter are electrically excited machines without permanent magnets. However, differing from the principle and function of the main motor, the exciter works as the rotating transformer to support the controlled field power for the main motor and, in this process, the rotating rectifier converts the AC current to DC current as a connecter between the exciter and the main motor. In light of this, the exciter and rotating rectifier form the brushless exciting system for the WFSM in EVs.

In Figure 2, the configuration of the WFSM drive system is depicted. In contrast with permanent magnetic excitation, the proposed WFSM is excited by current in the field windings, and the current is injected from the brushless exciting system in Figure 1. In terms of the principle of brushless excitation, the field windings of the main motor are excited via the proposed method called single phase AC excitation, which means that the field power converter supplies AC current $I_{ef}$ to the field windings of the exciter stator, and then the induced AC currents $I_{exa}$, $I_{exb}$, and $I_{exc}$ in the armature windings of the exciter rotor flow to the rotating rectifier, and finally the AC current $I_{ex}$ is converted to DC current $I_F$ for the field excitation of WFSMs. For the coordination with $I_F$, the currents $I_A$, $I_B$, and $I_C$ are regulated via a power converter and injected into armature windings to generate armature reaction magnetic potential. Therefore, under the joint action of exciting magnetic potential

and armature reaction magnetic potential, the torque of WFSM is generated, and it drives the shaft of EVs.

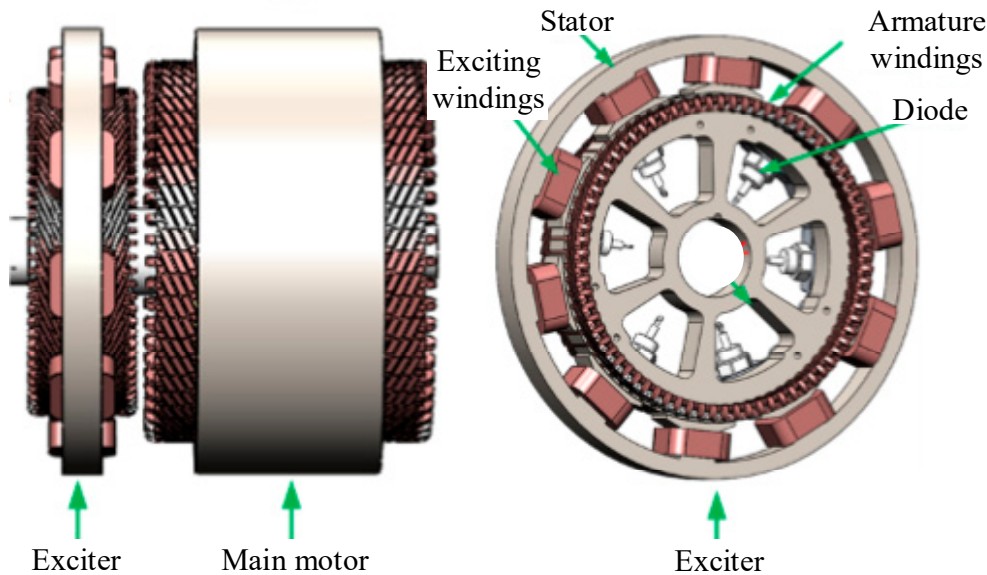

**Figure 1.** The two-stage structure of the proposed WFSM.

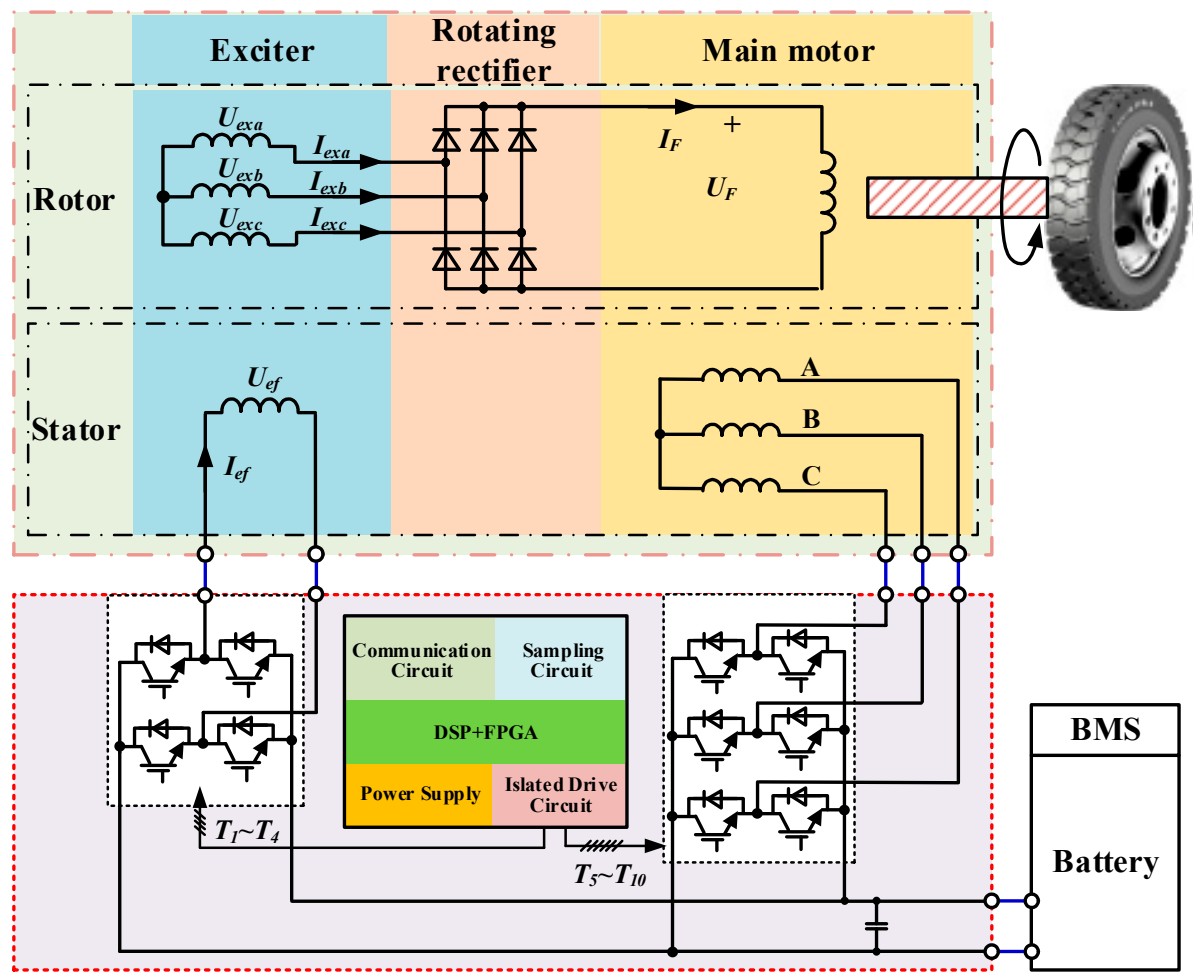

**Figure 2.** The configuration of the WFSM drive system.

According to the configuration and working principle of the WFSM, the power electronic converter is divided into two parts: the field power converter and the three-phase inverter. Two-level full-bridge topology is applied to the three-phase inverter, which is suitable for medium- and low-voltage applications. H-full bridge topology is applied to the field power converter and shares the DC bus with the three-phase inverter. The converter injects sinusoidal excitation current into the stator winding of the exciter, and the WFSM field is controlled by changing the current amplitude.

## 3. System Analysis and Development

### 3.1. Investigation and Analysis of the Main Motor

Figure 3 shows the 1/3 finite element analysis (FEA) model of the proposed WFSM, and the number of pole pairs is three. Based on this model, some motor characteristics are analyzed in this section.

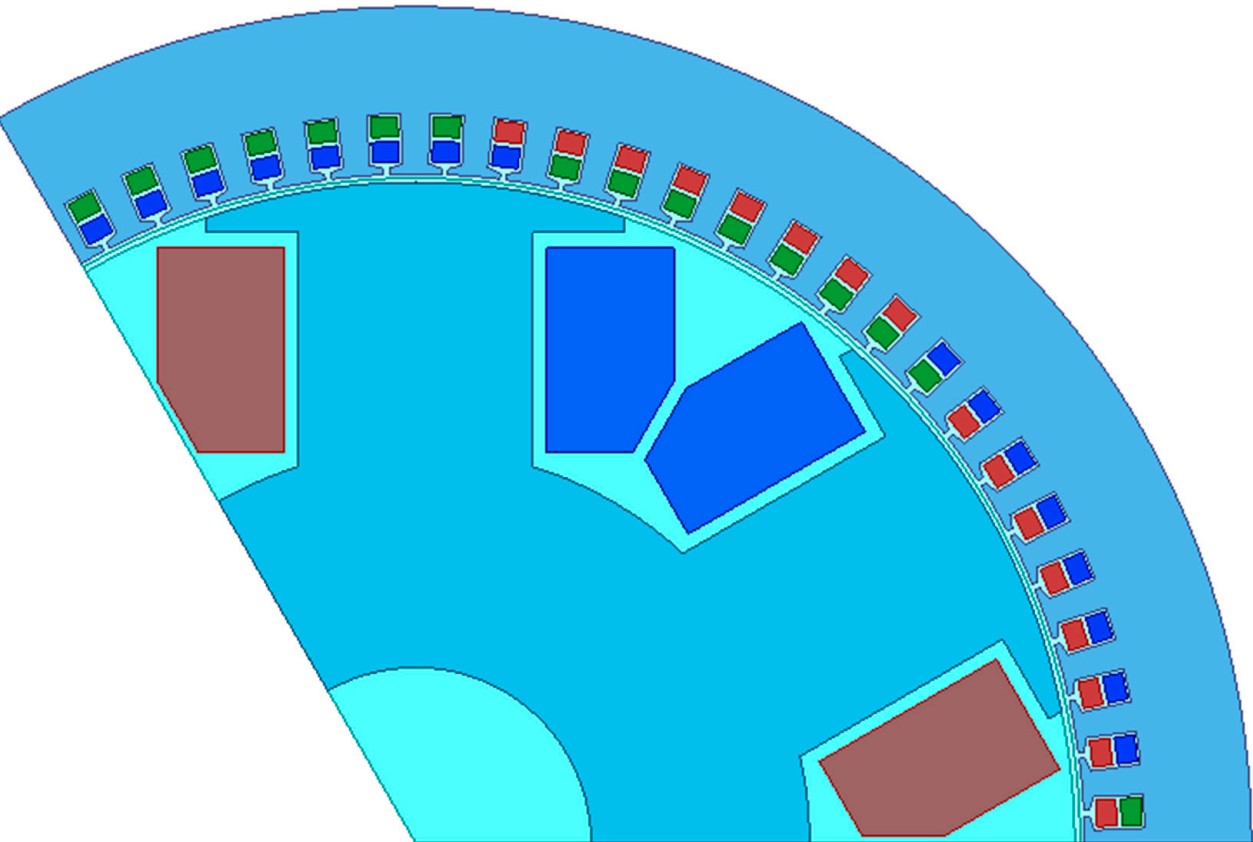

**Figure 3.** The 1/3 model of the main motor.

The Torque Characteristics of the Proposed WFSM

In Figure 4, the torque of the WFSM changing with the armature current under different exciting currents is depicted. In the condition of the constant exciting current, the torque $T$ is proportional to the armature current $I_{rms}$. Additionally, when the armature current $I_{rms}$ remains constant, the torque $T$ also increases with the exciting current. However, the torque $T$ has little change with an increase in exciting current $I_F$ when the excitation current exceeds 50 A, and the relationship between the torque $T$ and armature current $I_{rms}$ is approximate linearity. In fact, the proposed WFSM is designed under the rated working condition of exciting current at 60 A and armature current at 260 A. The nearly saturated core of the WFSM causes the linear characteristic of the relationship between torque $T$ and armature current $I_{rms}$.

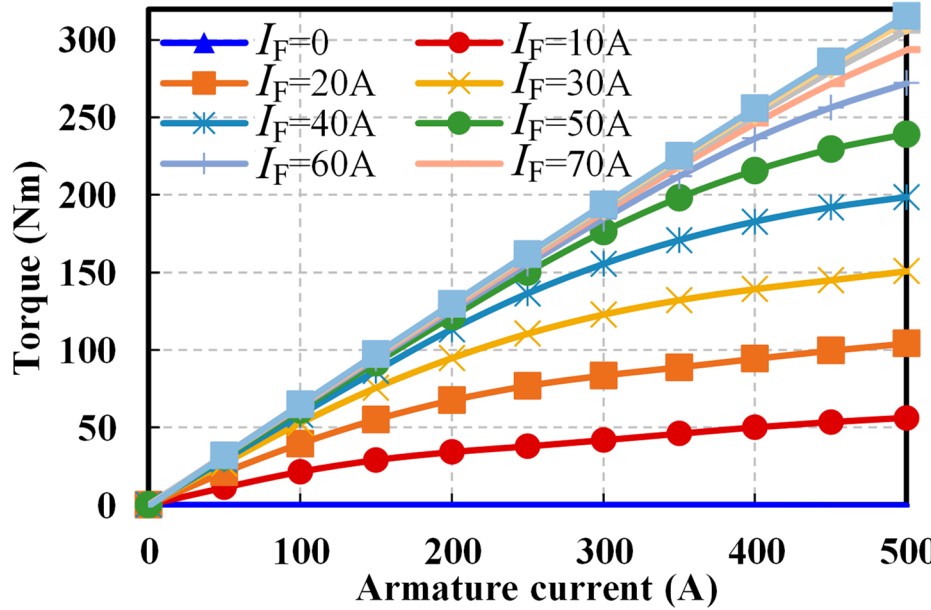

**Figure 4.** The torque of the WFSM changing with armature current under different exciting currents.

Figure 5 depicts the efficiency map of the WFSM. It only considers the electromagnetic loss of the WFSM, and some losses of mechanical aspects are ignored. As shown in Figure 5, the highest efficiency is up to 97% and the largest torque is 160 Nm. Over the wide range of the speed, the efficiency is more than 90%. At the rated speed of 8000 r/min, when the torque is the largest at 160 Nm, the efficiency of the proposed WFSM is still up to 97%. Additionally, the inductive line voltage is 306 V, which is lower than the DC bus voltage.

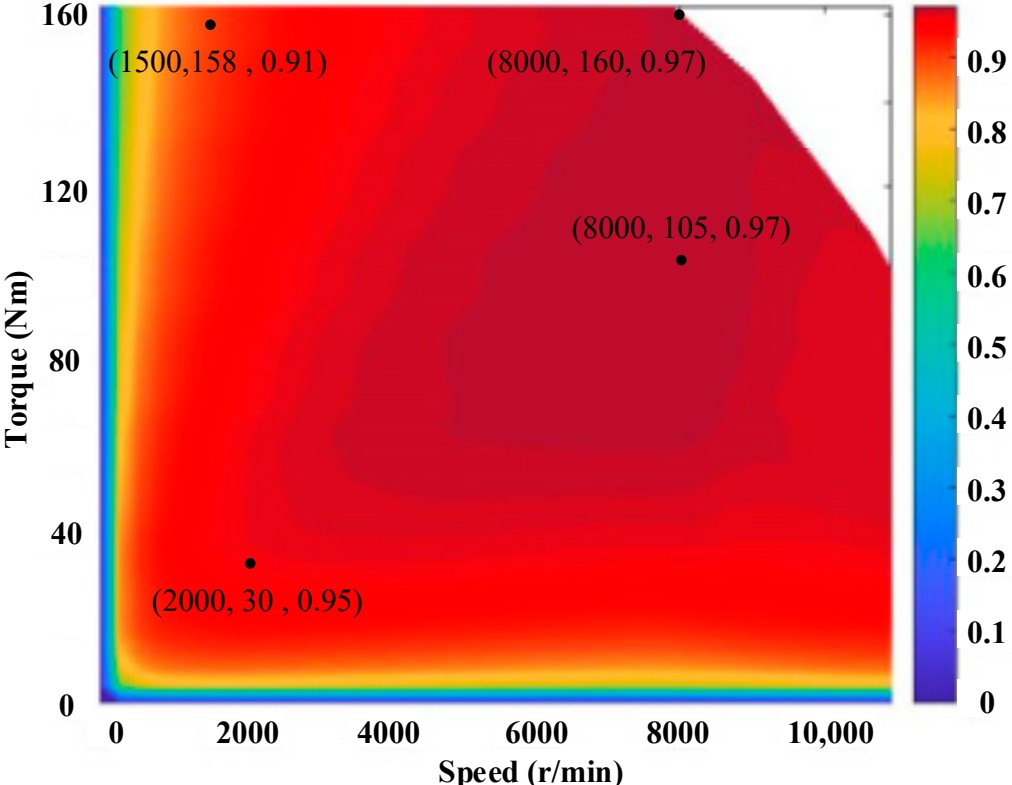

**Figure 5.** The efficiency map of the WFSM.

### 3.2. Analysis of the Exciter Characteristics

The 1/5 Maxwell model of the proposed exciter is shown in Figure 6. The single-phase centralized winding is wounded on salient poles of the stator for single-phase AC excitation. The three-phase distributed winding is set in the slots of the cylindrical rotor and generates inductive current for the excitation of the WFSM under the condition of single-phase AC excitation.

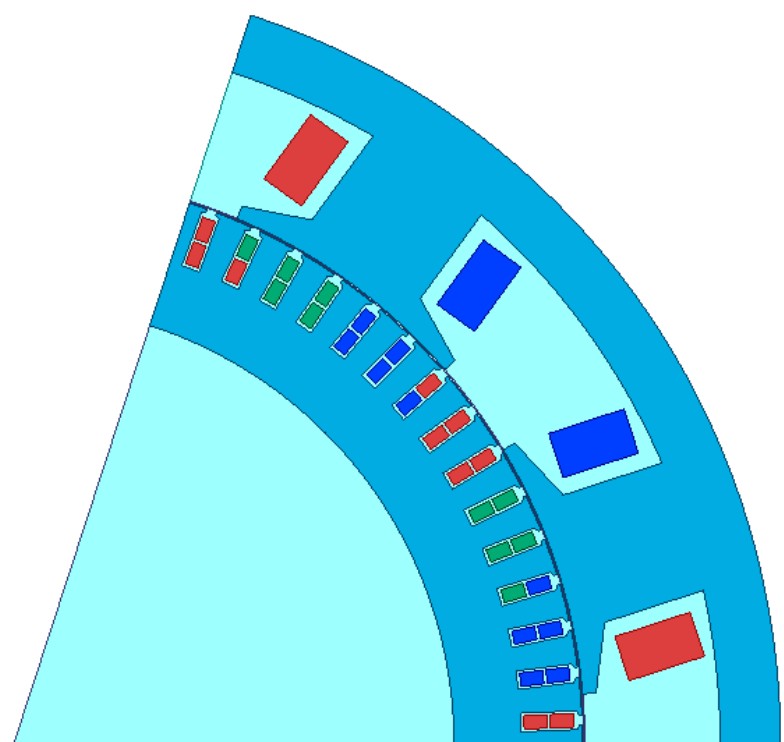

**Figure 6.** The 1/5 model of the exciter.

### 3.2.1. The Principle of Single-Phase AC Excitation

Figure 7 depicts the equivalent circuit of single-phase AC excitation, and the operation principle of the exciter is similar to the rotating transformer. The stator of the exciter works as the primary side of the transformer, and the rotor is the secondary side. When single-phase AC voltage is applied to the exciting winding, the pulsating magnetic field is generated in the stator, whose axis is on the center line of the stator pole, and is then coupled to the rotor. For this, the exciting power transmission is realized through the air gap between the stator and the rotor.

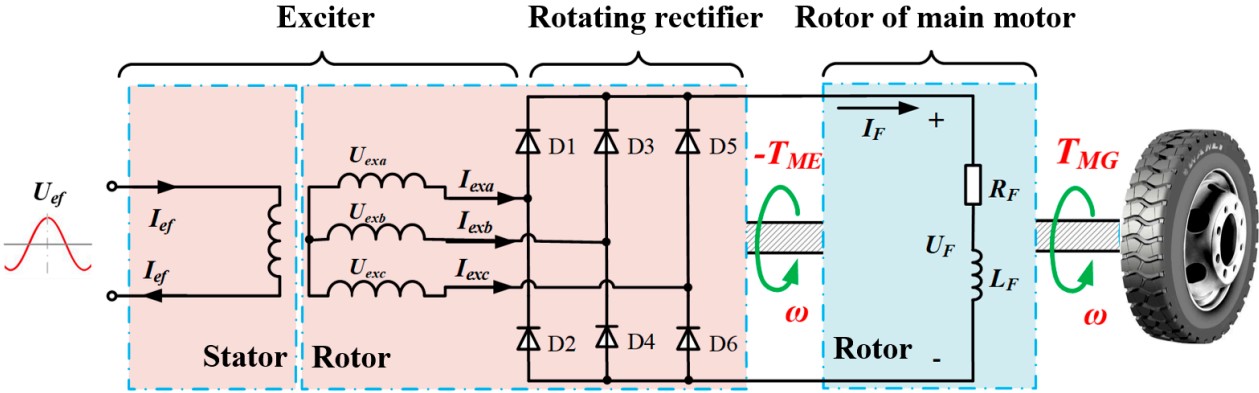

**Figure 7.** Equivalent circuit of single-phase AC excitation.

Figure 8 shows the induced electromotive force (EMF) of the exciter armature winding under different rotor position angles. Due to the rotation of the exciter rotor, the angle between the axis of the armature winding and the axis of the excitation winding keeps changing, and thus the EMF of the armature winding changes synchronously.

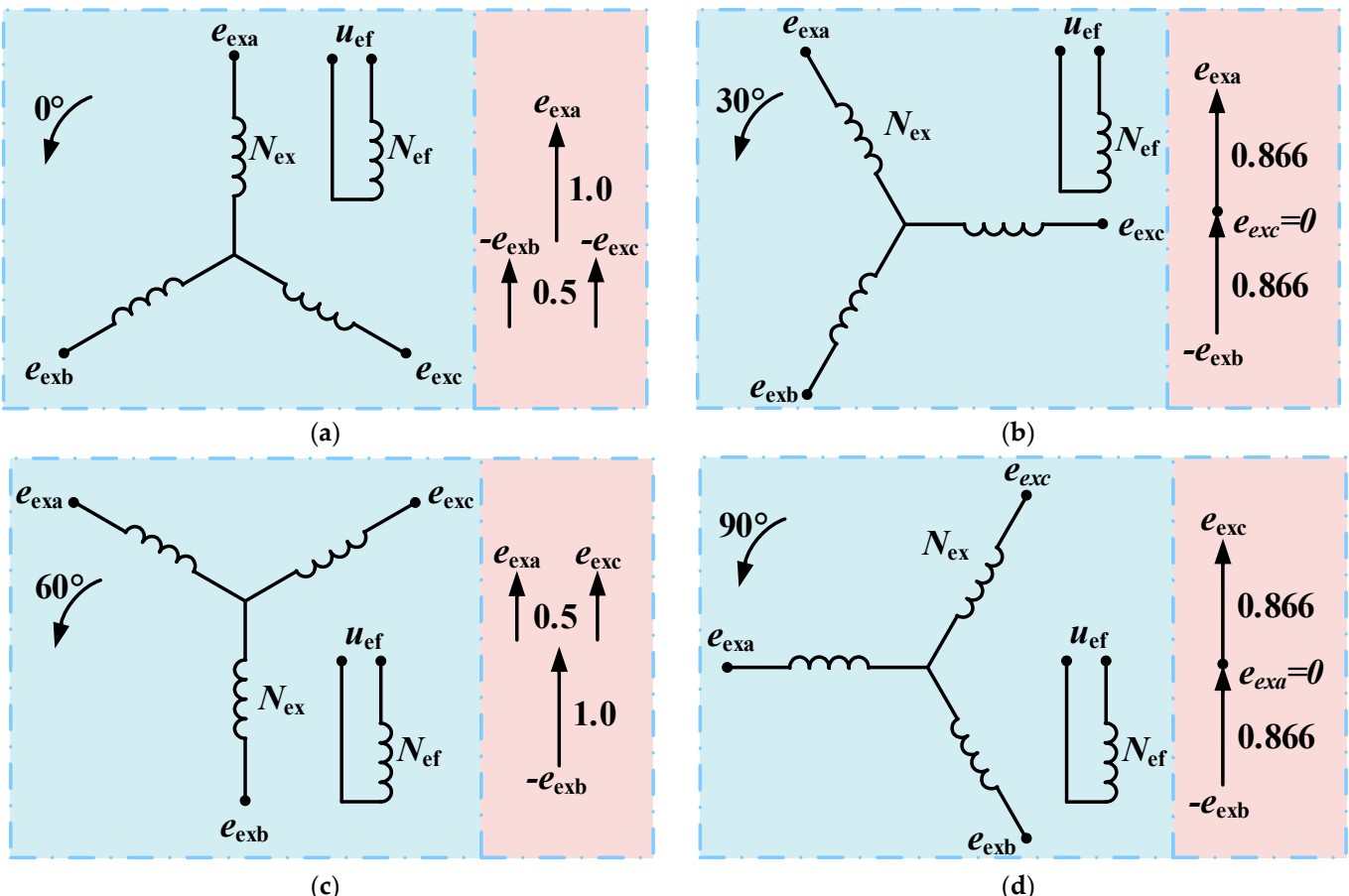

**Figure 8.** The EMF of exciter armature winding under different rotor position angles. (**a**) The angle between the phase A winding axis and the exciting winding axis is $0°$; (**b**) the angle between the phase A winding axis and the exciting winding axis is $30°$; (**c**) the angle between the phase A winding axis and the exciting winding axis is $60°$; (**d**) the angle between the phase A winding axis and the exciting winding axis is $90°$.

In Figure 8a, when the axis of phase A coincides with the axis of the exciting winding, the EMF of phase A is expressed as follows:

$$e_{exa} = \frac{k_{ex} N_{ex}}{N_{ef}} u_{ef}, \tag{1}$$

where $N_{ex}$ is the number of series turns per phase of exciter armature winding; $K_{ex}$ is the coefficient of armature winding; $N_{ef}$ is the number of series turns of exciting winding; and $U_{ef}$ is the exciting voltage.

Since the electrical angle between phase A, phase B, and phase C is $120°$, the EMF of phase B and phase C is as follows:

$$\begin{cases} e_{exb} = \frac{k_{ex} N_{ex}}{N_{ef}} u_{ef} \cos 120° = -\frac{k_{ex} N_{ex}}{2N_{ef}} u_{ef} \\ e_{exc} = e_{exb} \end{cases} \tag{2}$$

Then, the line voltage applied to the rotating rectifier is written as follows:

$$e_{exab} = e_{exac} = e_{exa} - e_{exb} = 1.5\frac{k_{ex}N_{ex}}{N_{ef}}u_{ef} \tag{3}$$

As shown in Figure 8b, due to a 30° electrical angle between the axis of phase A and the axis of the exciting winding, the EMF of phase A is described as follows:

$$e_{exa} = \frac{k_{ex}N_{ex}}{N_{ef}}u_{ef}\cos 30° = \frac{\sqrt{3}k_{ex}N_{ex}}{2N_{ef}} \tag{4}$$

The EMFs of phase B and C, respectively, are as follows:

$$e_{exb} = \frac{k_{ex}N_{ex}}{N_{ef}}u_{ef}\cos 150° = -\frac{\sqrt{3}k_{ex}N_{ex}}{2N_{ef}}, \tag{5}$$

$$e_{exc} = \frac{k_{ex}N_{ex}}{N_{ef}}u_{ef}\cos 270° = 0 \tag{6}$$

Furthermore, the line voltage applied to the rotating rectifier is as follows:

$$e_{exab} = e_{exa} - e_{exb} = 1.732\frac{k_{ex}N_{ex}}{N_{ef}}u_{ef} \tag{7}$$

In Figure 8c,d, the coupling relationship between armature winding and exciting winding is similar to that shown in Figure 8a,b, respectively. More specifically, only the phase changes, but the line voltage applied to the rotating rectifier is unchanged.

Therefore, the line voltage applied to the rotating rectifier is little changed as the rotation of the exciter rotor, and it can be expressed as the average value during rotation:

$$e_{exab} \approx 1.62\frac{k_{ex}N_{ex}}{N_{ef}}u_{ef} = 2.29\frac{k_{ex}N_{ex}}{N_{ef}}U_{ef}\sin(2\pi f_1 t), \tag{8}$$

where $U_{ef}$ is the RMS of the exciting voltage, and $f_1$ is the frequency of the exciting voltage.

According to Equation (8), it is concluded that the exciting current of WFSM, which is generated by the method of single-phase AC excitation, is nearly uninfluenced by the rotating speed. So, the $U_{ef}$ and the turn ratio $N_{ex}/N_{ef}$ can be increased to generate sufficient exciting current of the WFSM.

### 3.2.2. The FEA Analysis of Single-Phase AC Excitation

In order to further analyze the characteristics of single-phase AC excitation, the exciting current $I_{ef}$ of the exciter and the field current $I_F$ of the WFSM under different exciting frequency $I_F$ and rotating speed are simulated, based on the 1/5 Maxwell model of the exciter, which is in series with the diode rectifier and the exciting winding of the WFSM, as shown in Figure 7.

Figure 9 depicts the exciting currents $I_{ef}$ and $I_F$ changing with the exciting frequency $f_1$ under the condition of 0 r/min. With an increase in frequency $f_1$, the exciting current $I_F$ increases, but due to the low impedance caused by the low frequency, the exciting current $I_{ef}$ of the exciter is very large and reaches thousands of amperes, which is far beyond the bear of the winding. When the frequency $f_1$ is more than 300 Hz, the exciting current $I_{ef}$ and $I_F$ decrease with an increase in frequency. Additionally, in Figure 10, when $f_1$ is more than 400 Hz, the input apparent power $S_{ef}$ and output exciting power $P_{ef}$ gradually decrease, and the energy conversion efficiency $P_F/P_f$ is about 55%. Therefore, it is better for the exciting frequency $f_1$ to be at 400 Hz, considering the electric density of the exciting winding and the exciting power required by the WFSM.

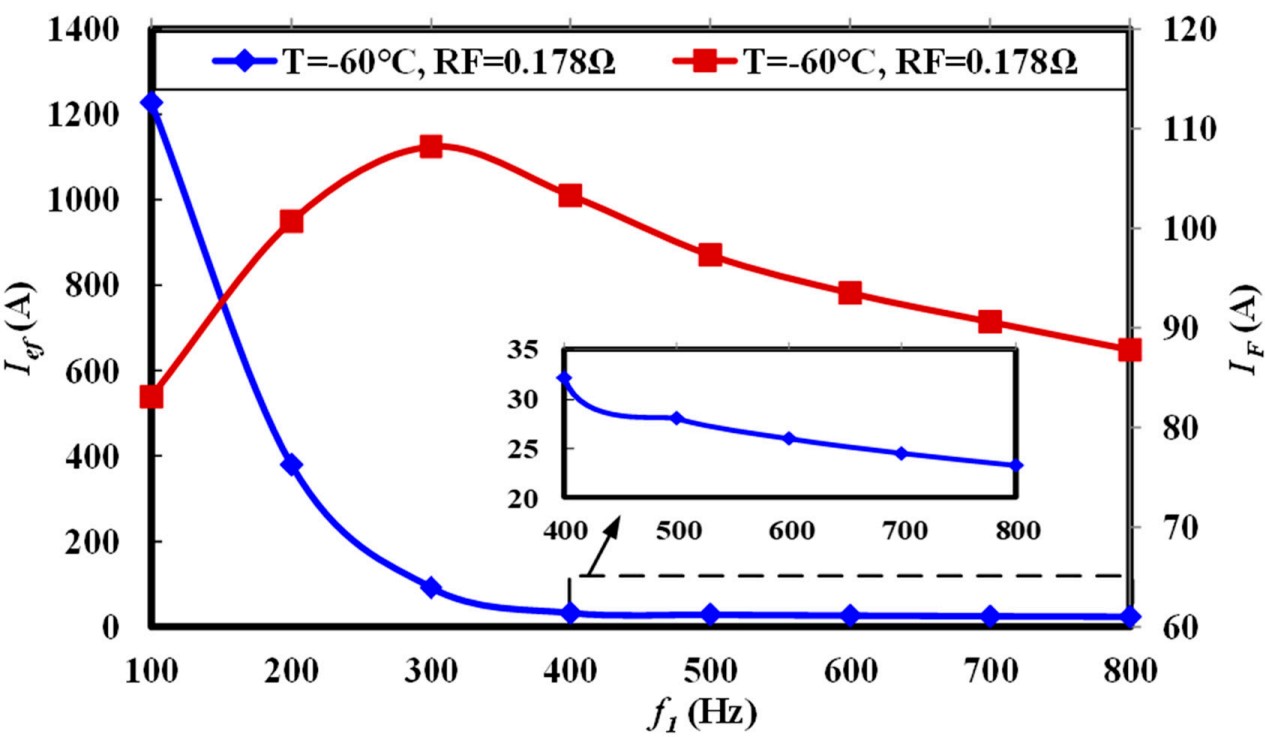

**Figure 9.** The curve of exciting current $I_{ef}$ and $I_F$ changing with exciting frequency $f_1$.

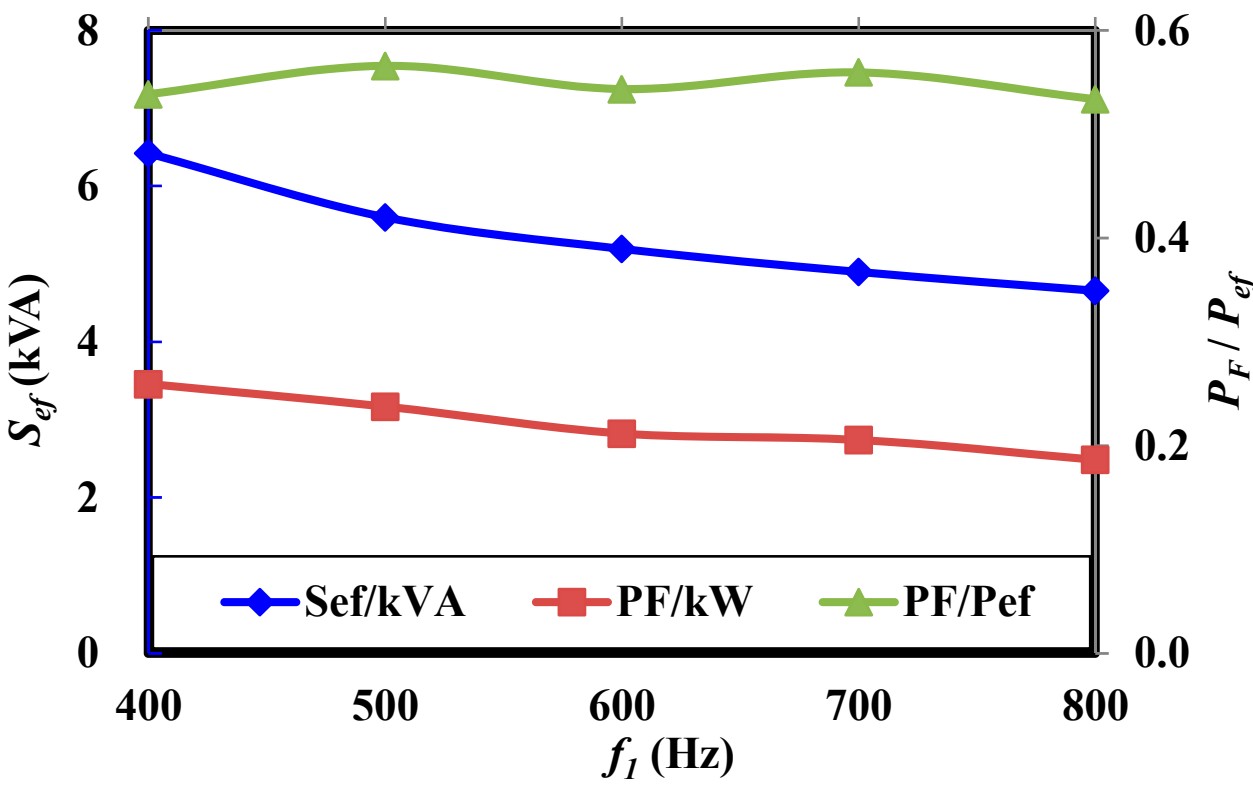

**Figure 10.** The curve of input apparent power $S_{ef}$, output power $P_F$, and transformation efficiency changing with frequency.

Considering the extreme weather conditions of electric vehicles and the overload condition of motors, Figure 11 shows the curve of exciting current $I_{ef}$ and the field current $I$ changing with rotating speed at −60 °C and 180 °C. It is depicted that under the given

temperature or field resistance RF, the exciting currents $I_{ef}$ and $I_F$ change inconspicuously with an increase in rotating speed, and basically remain constant, which is consistent with Equation (8). Even if the load resistance of the exciter increases by 2.36 times with a temperature increase from $-60\ ^\circ$C to 180 $^\circ$C, the changes in the exciting current $I_{ef}$ and $I_F$ are not obvious, which increase by less than 20%.

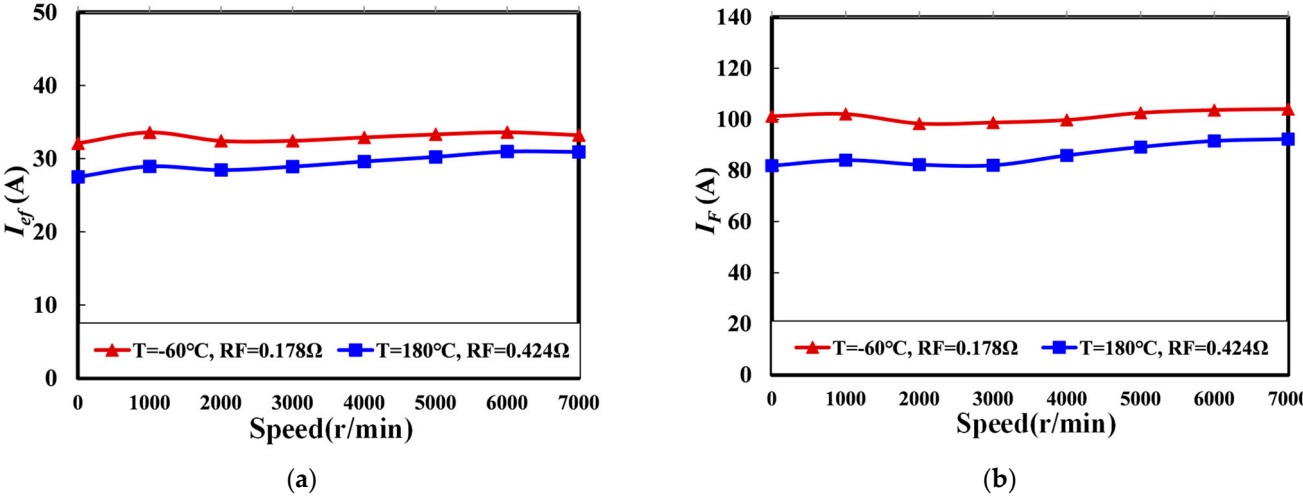

(**a**) (**b**)

**Figure 11.** The curve of $I_{ef}$ and $I_F$ changing with rotating speed under $-60\ ^\circ$C and 180 $^\circ$C: (**a**) the curve of exciting current $I_{ef}$; (**b**) the curve of exciting current $I_F$.

According to Figure 11, it is concluded that the exciter has the characteristic of the constant current source, which is not influenced by temperature and rotating speed. This is because the magnetic field of the exciter is unsaturated under any working condition, and the exciter always works in the linear region of the B-H curve.

Although the exciter works as a transformer, the magnetic field is partially skewed and torque will also be generated due to the action of armature reaction. Figure 12 shows the torque of the exciter changing with the speed. With an increase in speed, the resistant torque of the exciter gradually increases. However, it has little impact on the output torque of the WFSM, due to its small value.

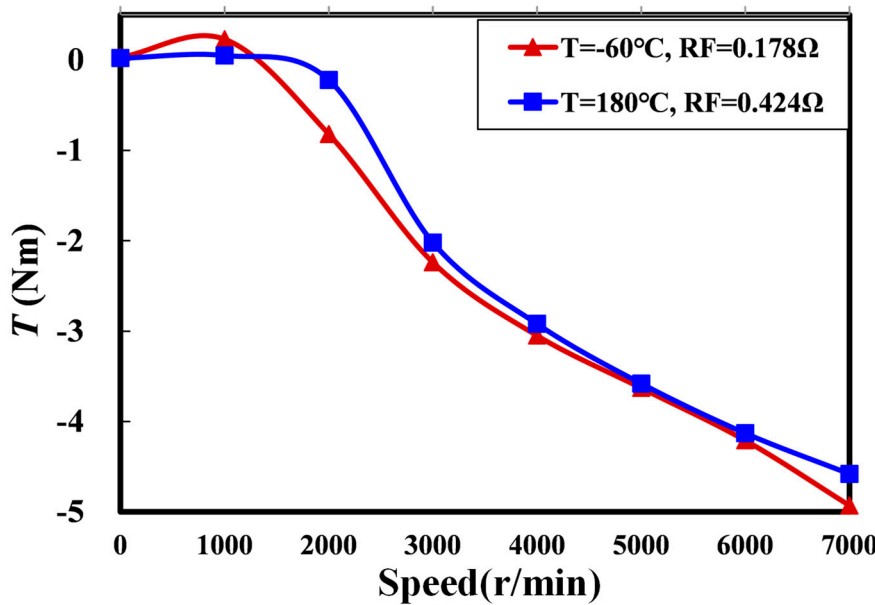

**Figure 12.** The curve of the torque of the exciter changing with the speed.

*3.3. Control Strategy of WFSM Drive System*

Similarly to the converter shown in Figure 2, the current regulator needs to be designed for armature control and excitation control, respectively. In this section, the design process of the regulator in the discrete domain is described in detail, and the coordinated control method is proposed to make the two parts no longer independent of each other.

3.3.1. Approaches of Discrete Time Armature Current Regulator Design

In the application of electric vehicles, the drive motor usually adopts the maximum torque per ampere (MTPA) control strategy [23] in order to avoid excessive power electronic converter size and reduce the cost per ampere of the system. The target torque of the drive motor is given by the electronic throttle, and the d–q axis current reference corresponding to the target torque is calculated through the MTPA table. It is noticeable that an important feature of the WFSM is that the *d*-axis inductance is greater than the *q*-axis inductance, which is different from the PMSM. Therefore, in order to improve the current torque per ampere, the WFSM usually operates in the first quadrant. The vector diagram is shown in Figure 13. Another important feature of the WFSM is that the d–q axis inductance is affected by magnetic circuit saturation, so the inductance will change with the field current, especially under high electromagnetic load. Therefore, different field currents will change the MTPA operating point, and the corresponding d–q axis current reference needs to be calculated according to the target torque and the field current, as shown in Figure 14. A three-dimensional MTPA table containing torque–excitation current–armature current data is obtained through pre-calibration.

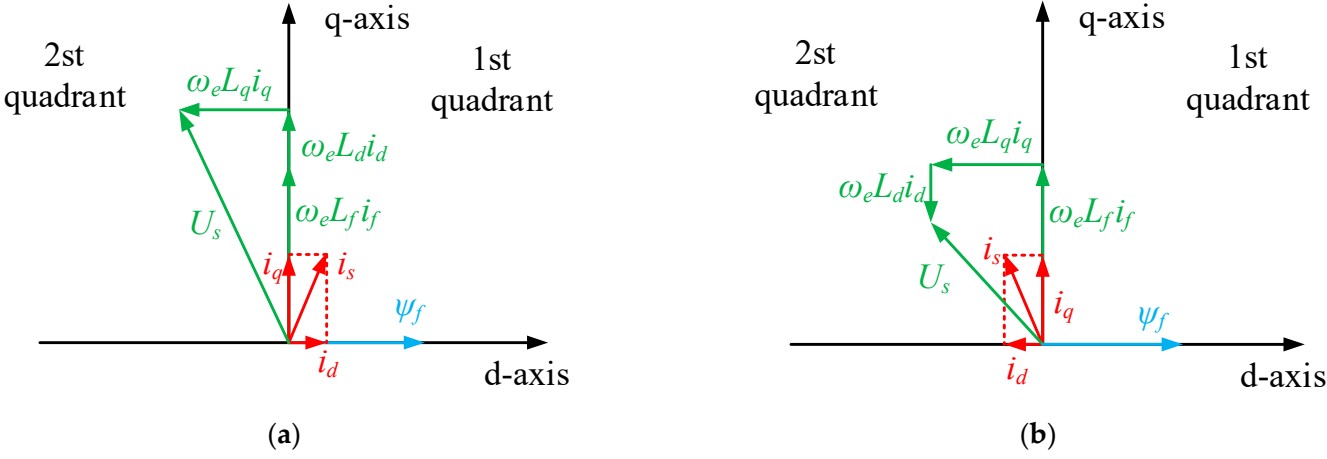

(**a**)　　　　　　　　　　　　　　　　　　　　(**b**)

**Figure 13.** Vector diagrams of the 1st and 2nd quadrant operation: (**a**) 1st quadrant operation; (**b**) 2nd quadrant operation.

The d–q axis current regulator is the key factor that affects the torque response ability of the drive motor. The widely used PI current regulator lacks clear parameter tuning principles, and it is difficult to achieve satisfactory results in situations with high control performance requirements. This paper applies the discrete time complex vector current regulator to the WFSM drive system, and designs the complex vector axis current regulator based on the discrete time model of the synchronous motor, which can directly obtain the discrete expression of the regulator and significantly improve its performance in the full-speed domain. The regulator parameters can be tuned directly according to the nominal motor model, and the gain can be adjusted according to the system stability and rapidity balance, which greatly shortens the development process of the control system.

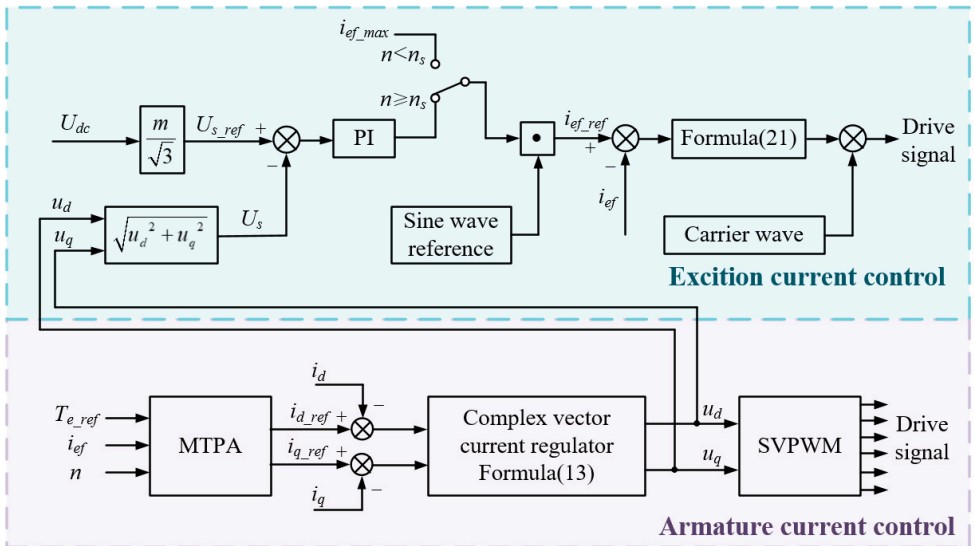

**Figure 14.** The block diagram of the current control system.

In the synchronous reference frame, the WFSM is modeled as

$$\begin{bmatrix} u_d \\ u_q \end{bmatrix} = \begin{bmatrix} R_s + pL_d & -\omega_e L_q \\ \omega_e L_d & R_s + pL_q \end{bmatrix} \begin{bmatrix} i_d \\ i_q \end{bmatrix} + \begin{bmatrix} 0 \\ \omega_e \psi_f \end{bmatrix}, \tag{9}$$

where $u_d$, $u_q$, $i_d$, $i_q$, $R_s$, $\omega_e$, $L_d$, $L_q$, $\psi_f$, and $p$ denote the d–q axis stator voltage, the d–q axis stator current, the stator resistor, the motor electrical angular frequency, the d–q axis stator inductance, the flux linkage due to the field current, and the differential operator, respectively.

The classical transformation of the physical system in Equation (9) to the discrete time domain can be accomplished by modeling the inverter as a unity gain ideal zero-order hold voltage latch. The transfer function is discretized using the method in the previous research [24]:

$$G_{plant}(z) = \frac{i_{dq}(z)}{u_{dq}(z)} = \frac{e^{-j\omega_e T_s} - e^{-\frac{R_s + j\omega_e L_q}{L_d} T_s}}{z e^{j\omega_e T_s} [R_s - j\omega_e (L_d - L_q)] (z - e^{-\frac{R_s + j\omega_e L_q}{L_d} T_s})}, \tag{10}$$

where $T_s$ denotes the sampling period and the control computation and the PWM update delay is modeled as $1/ze^{j\omega_e T_s}$.

Due to the difference in the d–q axis inductance of WFSM, the current regulator designed according to the transfer function in Equation (10) is very complex. In the case of no obvious performance loss, the normalized $L_s$ is usually used to replace $L_d$ and $L_q$ for modeling:

$$L_s = (L_d + L_q)/2 \tag{11}$$

The discrete time domain transfer function of the WFSM can be simplified as

$$G_{plant}(z) = \frac{i_{dq}(z)}{u_{dq}(z)} = \frac{e^{-j\omega_e T_s} - e^{-\frac{R_s + j\omega_e L_s}{L_s} T_s}}{z e^{j\omega_e T_s} R_s (z - e^{-\frac{R_s + j\omega_e L_s}{L_s} T_s})} \tag{12}$$

In order to analyze the performance of the current regulator separately, the influence of control delay is not considered in this section. The zero-pole cancellation principle determines that the zero of the regulator is the same as the pole of the plant. According to the internal model principle, the denominator of the regulator transfer function is selected

as the z-domain expression of the integrator. The current regulator is designed in the form of

$$G_c(z) = \frac{K_{dq}(e^{j\omega_e T_s} - z^{-1}e^{-\frac{R_s}{L_s}T_s})}{1 - z^{-1}}$$

$$K_{dq} = K\frac{R_s}{1 - e^{-\frac{R_s}{L_s}T_s}},$$

(13)

where $K$ is the integral gain of the axis, and the response performance of the current loop can be easily changed by adjusting the $K$.

The amplitude–frequency response of $G_{plant}$ and $G_{op}$ is shown in Figure 15, where $K = 0.35$. After the compensation of the current regulator $G_c$, the system has a sufficient gain margin and phase margin, and the low-frequency gain is also raised by the integrator. The performance of the current loop compensated by $G_c$ is satisfactory.

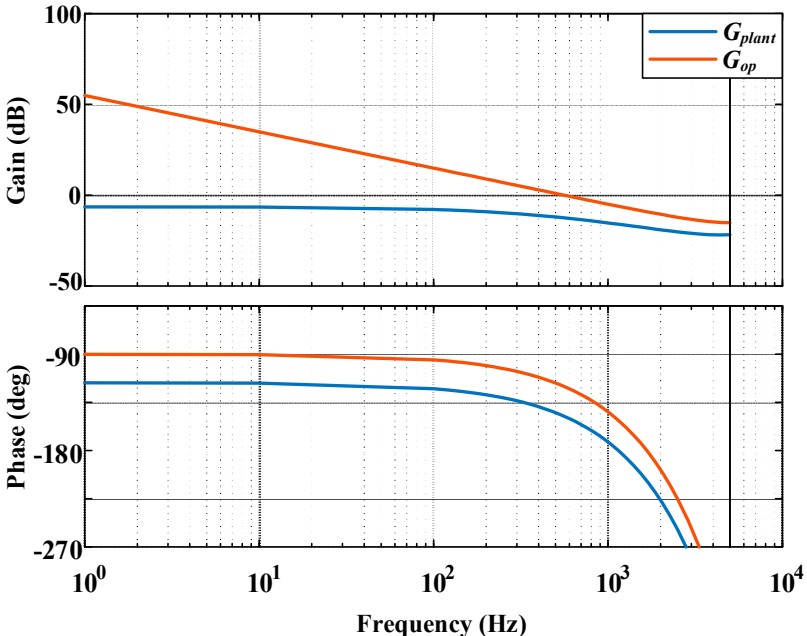

**Figure 15.** The bode diagram of $G_{plant}$ and $G_{op}$.

Because accurate motor parameters are difficult to obtain and are greatly affected by temperature and working conditions, it is necessary to analyze the performance of the current regulator in the case of mismatched parameters, which are divided into resistance mismatch and inductance mismatch.

Figure 16 shows that the performance of the current loop is more sensitive to the inductance parameters. The change in resistance only slightly changes the phase–frequency curve, and the change in the phase margin can be ignored. However, the influence of inductance on the amplitude–frequency curve is noteworthy. The decrease in inductance caused by magnetic circuit saturation will lead to a significant increase in the crossing frequency. The reduction in the gain margin can improve the dynamic response of the current loop to a certain extent, but it may also lead to critical stability or even instability of the system when the regulator parameters are large. Considering the stability effect caused by parameter perturbation, one can yield the range of $K$ as

$$K \in (0.15, 0.35)$$

(14)

### 3.3.2. Design of Excitation Current Coordination Control Strategy

The field power converter injects 400 Hz excitation current into the rotating transformer, and the current amplitude affects the field of WFSM. The converter needs to

provide the excitation current with the correct amplitude in the full speed range. This section proposes an excitation current control method based on the constant voltage vector.

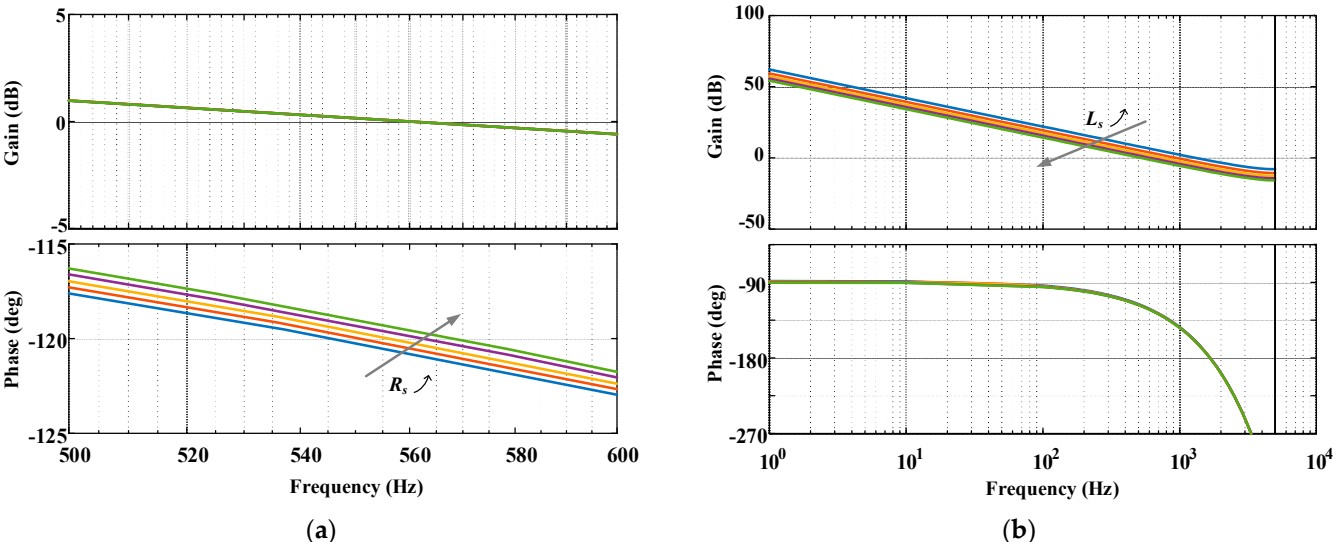

**Figure 16.** The bode diagram of parameters' mismatch: (**a**) resistance changes from $0.5R_s$ to $3R_s$; (**b**) inductance changes from $0.5L_s$ to $2L_s$.

The voltage vector $U_s$ applied to the WFSM armature through the inverter is as follows:

$$\vec{U_s} = u_d + ju_q \tag{15}$$

The voltage vector amplitude output by the voltage source inverter is clamped by the DC bus voltage $U_{dc}$:

$$\left| \vec{U_s} \right| = \sqrt{u_d^2 + u_q^2} = m\frac{U_{dc}}{\sqrt{3}} \;\; m < 1, \tag{16}$$

where $m$ denotes the modulation ratio. In order to improve the utilization of the DC bus voltage as much as possible, $m$ is usually taken to be about 0.9.

The voltage vector satisfies the following voltage balance equation:

$$\vec{U_s} = \vec{E} + R_s\,\vec{i} + j\omega_e L_d i_d + j\omega_e L_q i_q \tag{17}$$

The amplitude of the WFSM back EMF $E$ can be adjusted by the field current, and the voltage vector $U_s$ can avoid being clamped by the DC bus voltage during high-speed operation, which is the key factor for the WFSM to realize constant power operation in a wider speed range. Therefore, the excitation current amplitude reference can be calculated by selecting the appropriate voltage vector amplitude, and the voltage vector amplitude can be kept constant through closed-loop control. The control block diagram is shown in Figure 14. In the low speed range, the excitation current will always maintain the upper limit to obtain the maximum torque per ampere. After entering the high speed range, the excitation current amplitude reference is calculated by the PI regulator. Note that its reference needs to be limited to avoid magnetic circuit saturation.

The excitation current of the rotating transformer is 400 Hz constant frequency AC. The proportional resonance (PR) controller is widely used in constant frequency situations. The transfer function of the PR controller is expressed as follows:

$$G_{PR}(s) = K_{\mathrm{p}} + \frac{2K_{\mathrm{r}}\omega_{\mathrm{c}}s}{s^2 + 2\omega_{\mathrm{c}}s + \omega_{\mathrm{o}}^2} \tag{18}$$

where $K_p$, $K_r$, $\omega_o$, and $\omega_c$ denote the proportional coefficient, the resonance coefficient, the resonant angular frequency, and the resonant bandwidth angular frequency. The transfer function in the s-domain is mapped to the z-domain through bilinear transformation:

$$s = \frac{2}{T_s}\frac{z-1}{z+1}$$

$$G_{PR}(z) = \frac{4K_r\omega_c T_s(z-1)(z+1)}{4(z-1)^2+4\omega_c T_s(z^2-1)+\omega_o^2 T_s^2(z+1)^2} \tag{19}$$

The excitation inductance of the rotating transformer is large and the field power converter operates under hard switching, so the low switching frequency can be selected to reduce the switching loss of the converter. Therefore, the switching frequency of the converter is selected as 4 kHz and the sampling period $T_s$ is 0.25 ms. However, bilinear transformation is a type of nonlinear mapping. Mapping points in the left-hand plane of the s-domain to the unit circle of the z-domain will cause frequency distortion, which will increase with an increase in the $T_s$, as shown in Figure 17. Therefore, the resonance frequency designed in the s-domain will shift after being transformed into the z-domain, which needs to be corrected.

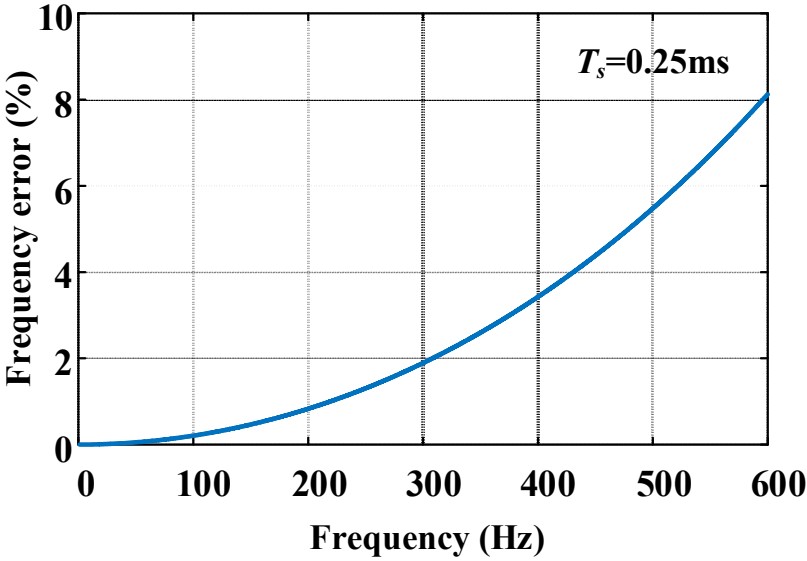

**Figure 17.** The frequency error of bilinear transformation.

Redefine the mapping as

$$s = K_c\frac{z-1}{z+1}, \tag{20}$$

where $K_c$ is the correction coefficient.

The z-domain transfer function of $G_{PR}$ can be expressed as

$$G_{PR}(z) = \frac{2K_r\omega_c K_c(z-1)(z+1)}{K_c^2(z-1)^2 + 2\omega_c K_c(z^2-1) + \omega_o^2(z+1)^2} \tag{21}$$

By solving the poles of Equation (21) and matching them with the designed resonant frequency, $K_c$ can be expressed as

$$K_c = \frac{\omega_o}{\tan(\frac{\omega_o T_s}{2})} \tag{22}$$

After frequency correction, the resonance frequency will return to the pre-designed value, as shown in Figure 18. Although this method will bring additional distortion at other frequencies, the impact on the system is acceptable due to its low gain.

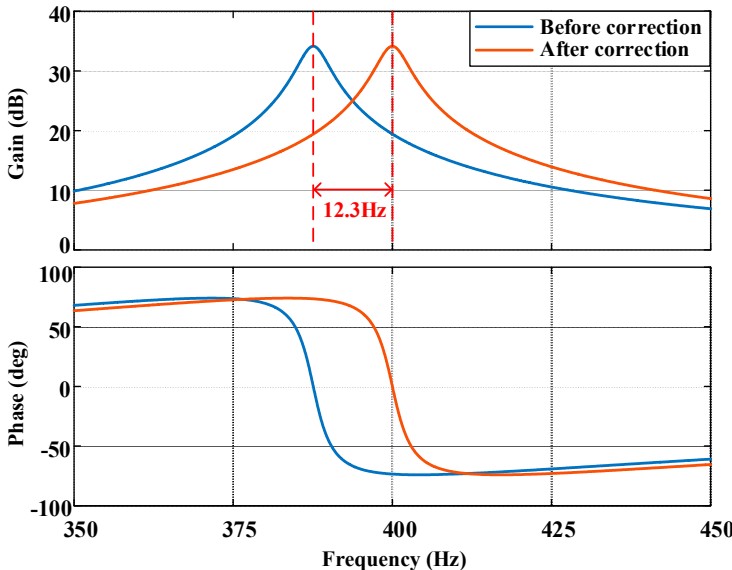

**Figure 18.** The bode diagram of $G_{PR}$ before and after correction.

## 4. Experimental Verification

In this section, the WFSM investigated in this paper is experimentally verified. The experimental platform is shown in Figure 19. The lengths of the stator laminations and armature winding end pairs for the motor are 100 mm and 25 mm, respectively. The lengths of the stator laminations and armature winding end pairs for the exciter are 28 mm and 15 mm, respectively. The windings and diodes in the WFSM rotor will generate a lot of heat. Therefore, a centrifugal fan is used to provide sufficient air pressure to ensure heat dissipation. The air inlet grid at the front of the car provides cooling air for the WFSM.

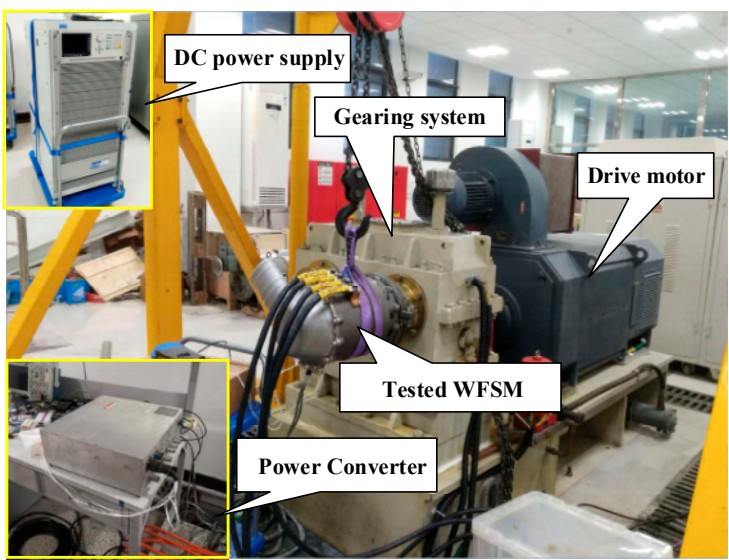

**Figure 19.** WFSM test platform.

During the experiment, the tested WFSM runs in the current control mode. The drive motor connected to the tested motor runs in the speed control mode and maintains constant speed of the tested WFSM.

### 4.1. AC Excitation Characteristics

During this experiment, in order to test the excitation characteristics of the WFSM and eliminate the influence of the control loop on the system, AC power supply is used to

excite the rotating transformer, as shown in Figure 20. The experimental results are shown in Figure 21. When different excitation voltages $U_{ef}$ are applied to the rotating transformer, the corresponding excitation current $I_{ef}$ remains constant at different speeds and increases linearly with $U_{ef}$.

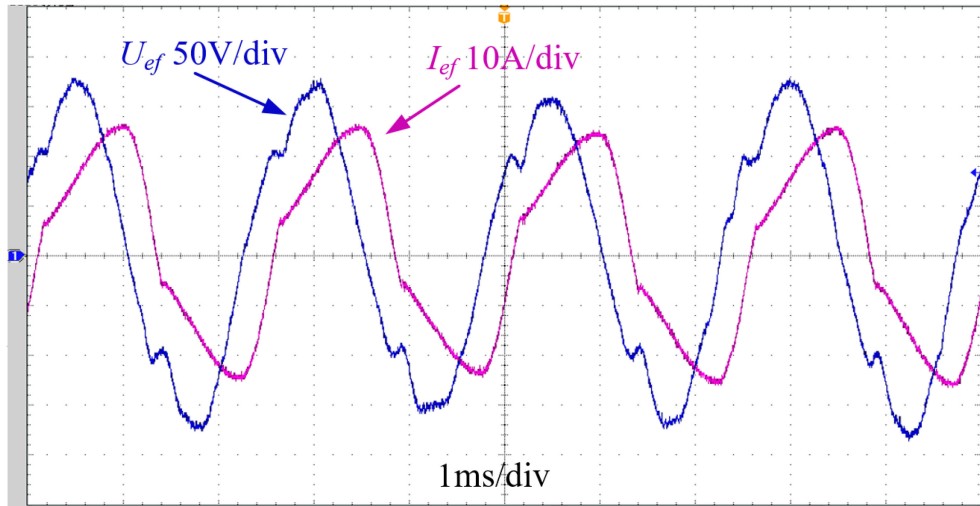

**Figure 20.** The waveform of AC excitation voltage $U_{ef}$ and current $I_{ef}$.

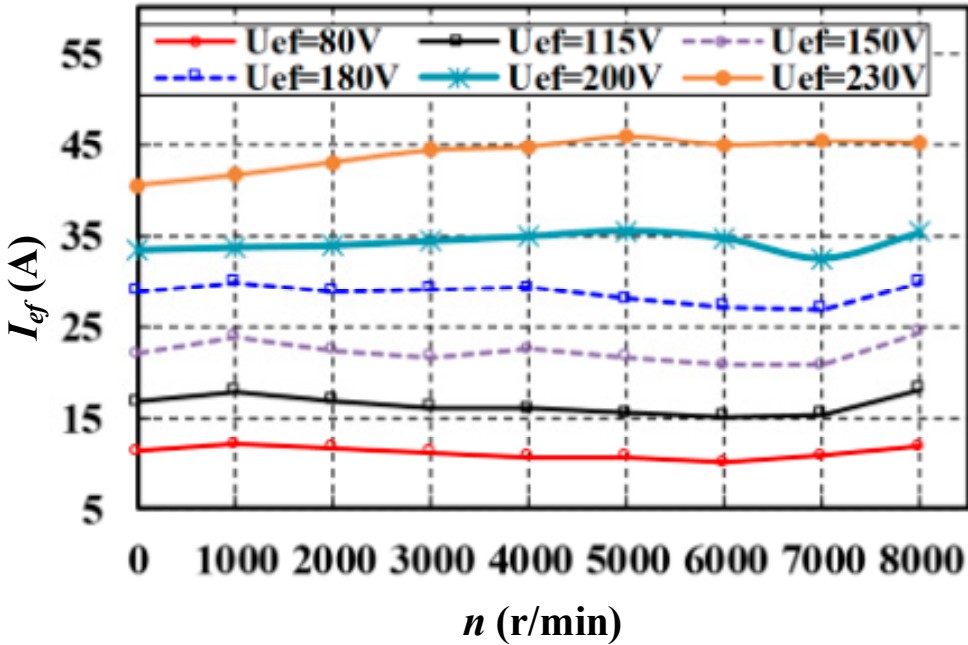

**Figure 21.** The curve of excitation current $I_{ef}$ changing with speed.

*4.2. Torque under Different Armature Current*

The torque output capability at a low speed is one of the important indicators to measure the performance of the drive motor. In order to discuss the relationship between current amplitude and torque separately, the influence of the current angle on torque is not considered. The $d$-axis current is set to 0, so the reluctance torque component of the WFSM can be ignored. It is hoped that the back EMF is high enough to reduce the output current of the three-phase inverter. The main motor needs a large enough field current to make the magnetic circuit close to saturation. According to the no-load back EMF characteristics obtained from the experiment, set the excitation current to 17.2 A and test the torque output using different current amplitudes in the low speed zone. Since it has been proved in the

previous section that the field current is independent of the speed, the relationship between the torque and current is consistent at different speeds. The experimental results are shown in Figure 22. It is proved that the designed WFSM has the same performance as expected. Additionally, the efficiency test results are shown in Figure 23.

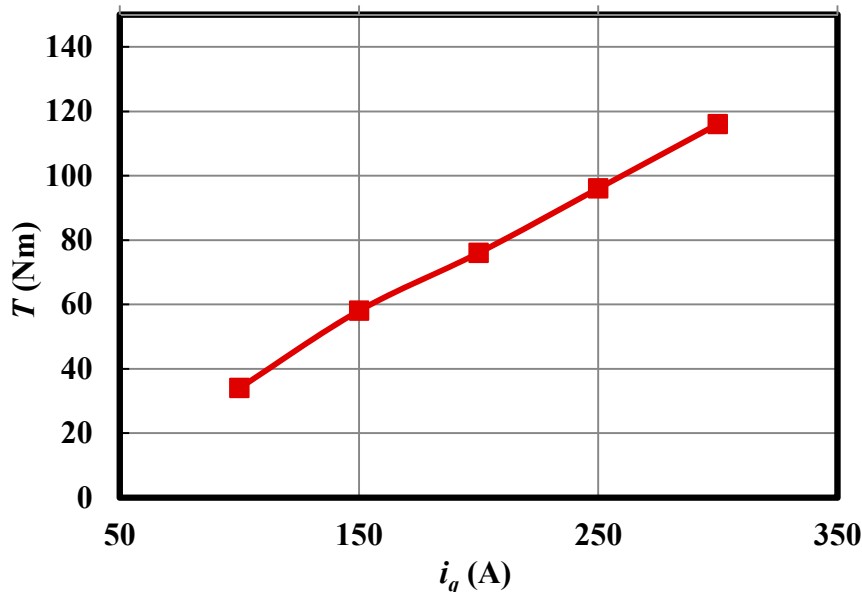

**Figure 22.** WFSM torque versus *q*-axis current $i_q$ at 2000 r/min.

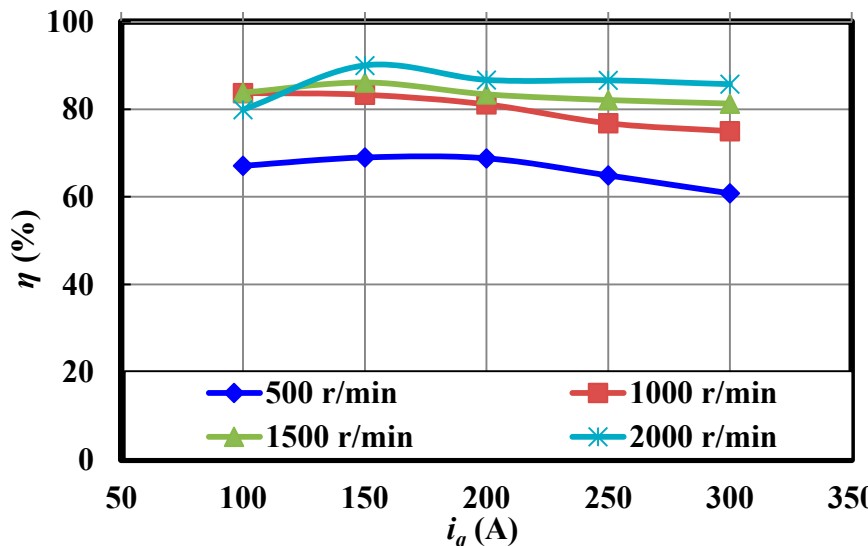

**Figure 23.** Efficiency versus current and speed (including three-phase inverter efficiency).

## 5. Conclusions

In this paper, to realize the brushless and magnetless operation, a two-stage WFSM is proposed for EV applications by replacing the permanent magnet with the rotor field windings while no brushes and slip rings are introduced. This is an environmentally friendly solution for electric drive components. The operation principle and characteristics of the proposed WFSM are analyzed in this paper. The experimental verification and analysis show that the WFSM has the following characteristics:

- The exciter excited by single-phase constant frequency current has the characteristics of current amplification and constant current source, and the amplification factor is almost independent of load and speed;

- The torque of the main motor is proportional to the armature current;
- The additional control DOF introduced by the field current is the key for WFSM to obtain high torque capability in the low speed region and to have a wider constant power region.

The coordinated control strategy of the armature current and excitation current proposed in this paper solves the control problem of the physical cross-coupling between the direct axis and the field axis. A high-performance torque response is achieved by the armature current and excitation current regulators based on the WFSM discrete time model.

**Author Contributions:** Conceptualization, Z.Z. and J.L.; methodology, Z.Z.; software, Y.L.; validation, Y.L., Y.W. and J.L.; formal analysis, Y.L. and Y.W.; investigation, Y.L. and Y.W.; resources, Z.Z. and J.L.; writing—original draft preparation, Y.L. and Y.W.; writing—review and editing, Z.Z. and J.L.; supervision, Z.Z.; project administration, J.L. All authors have read and agreed to the published version of the manuscript.

**Funding:** This research was funded by Jiangsu Provincial "333 Project" Funds for High-Level Talents, grant number BRA2020042.

**Data Availability Statement:** Not applicable.

**Conflicts of Interest:** The authors declare no conflict of interest.

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
