# Peer review of "Investigation and Development of the Brushless and Magnetless Wound Field Synchronous Motor Drive System for Electric Vehicle Application"

_wevj, doi:10.3390/wevj14040081_

Round 1

Reviewer 1 Report

The article discusses different aspects of designing an electric drive of an electric vehicle with a wound field synchronous motor (WFSM) exited by AC brushless exciter. An experimental verification of the obtained theoretical results is presented. The article may be of interest to researchers in the field of designing electrical machines. However, the following points need to be explained in more detail:

1) Figure 3 shows the starting/damper winding on the rotor. Why this winding for an inverter-fed motor?

2) Please provide information on the lengths of the stator laminations and armature winding end parts for the main motor and exciter.

3) Provide information about the efficiency of the exciter under various loading conditions.

4) Provide details of the rotor and exciter cooling system.

5) Please add a photo of the prototype exciter.

6) Please add a photo of the rotor of the prototype WFSM.

7) Many papers on traction WFSMs that the reviewer is aware of are not covered in the introduction. I think the introduction should be extended considerably.

8) What are the power rating and the current rating of the solid-state inverter to power the proposed WFSM? What inverter ratings are necessary for an induction motor in the same application?

Author Response

Dear professor:
Thanks for your letter and comments to our manuscript No. wevj-2279124 "Investigation and Development of the Brushless and Magnetless Wound Field Synchronous Motor Drive System for Electric Vehicle Application".
We would like to thank all the reviewers and AE for the valuable and constructive comments for improving the quality of our manuscript. We have studied the comments and suggestions carefully.

Please refer to the attachment "Response to Reviewer 1 Comments" for the reply to your comments, and the revised manuscript is also included.

We greatly appreciate your comments to our manuscript and look forward to receiving your further suggestions.

Thank you and best regards.
Yours sincerely.

Reviewer 2 Report

This paper develops a magnet less wound field synchronous motor (WFSM) drive system for electric vehicle application to solve the problem of soaring cost and supply fluctuation of permanent magnet materials. In general, this is a good paper dealing with the timely subject.

The following comments and recommendations may be helpful to improve the paper:

Q-1: In the introduction, it is better to introduce some advantages of motor drive-based electric vehicle, such as fast motor response control and wheel speed acquisition with some related work. This is beneficial for vehicle state estimation and underlying dynamics control: Automated vehicle sideslip angle estimation considering signal measurement characteristic; Vision‐aided intelligent vehicle sideslip angle estimation based on a dynamic model.

Meanwhile, due to several potential benefits, WFSM can also be applied in distributed drive electric vehicles, which needs to be discussedA hierarchical energy efficiency optimization control strategy for distributed drive electric vehicles; Comprehensive chassis control strategy of FWIC‐EV based on sliding mode control.

Q-2: In the modelling of motor, more physical characteristics in the real-world application of EVs should be considered, such as backlash or others features, and the physical compensation should be discussed: Research on synchronous control strategy of steer-by-wire system with dual steering actuator motors.

Q-3: In Fig.16, there are many different lines, so it is better to give some explanations for different lines. Further, the curve on the top left corner, do the two lines overlap exactly or is there only one line?

Q-4: When designing the control strategy of WFSM drive system, how do you identify these parameters for system transfer functions? 

Author Response

Dear professor:
Thanks for your letter and comments to our manuscript No. wevj-2279124 "Investigation and Development of the Brushless and Magnetless Wound Field Synchronous Motor Drive System for Electric Vehicle Application".
We would like to thank all the reviewers and AE for the valuable and constructive comments for improving the quality of our manuscript. We have studied the comments and suggestions carefully.

Please refer to the attachment "Response to Reviewer 2 Comments" for the reply to your comments, and the revised manuscript is also included.

We greatly appreciate your comments to our manuscript and look forward to receiving your further suggestions.

Thank you and best regards.
Yours sincerely.

Round 2

Reviewer 2 Report

I find that the authors have put considerable effort into addressing the comments of the reviewers. As a result, the paper is very much improved, and I have no problem recommending it for publication.